

# No tears in heaven: did the media create the pseudo-phenomenon "altitude-adjusted lachrymosity syndrome (AALS)"?

Paul Wicks[1] and Lee Lancashire[2]

[1] PatientsLikeMe, Cambridge, MA, United States of America
[2] Cohen Veterans Bioscience, Cambridge, MA, United States of America

## ABSTRACT

**Objective**. In the media, numerous public figures have reported involuntary emotional outbursts arising from watching films on planes, resembling neurological phenomena such as pseudobulbar affect. Putative risk factors put forward include altitude, mild hypoxia, or alcohol. Our objective was to determine whether watching a film on an airplane is really more likely to induce involuntary, uncontrollable, or surprising crying than watching one on the ground, described in some social media as "altitude-adjusted lachrymosity syndrome" (AALS), or whether this is a pseudo-phenomena.

**Methods**. Amazon Mechanical Turk survey participants ($N = 1,084$) living in the United States who had watched a film on a plane in the past 12 months were invited to complete an online survey. The main outcome measures were likelihood of crying in a logistic regression model including location of viewing, age, gender, genre of film, subjective film rating, annual household income, watching a "guilty pleasure" film, drinking alcohol, feeling tired or jetlagged, or having a recent emotional life event.

**Results**. About one in four films induced crying. Watching a film on a plane *per se* does not appear to induce involuntary crying. Significant predictors of crying included dramas or family films, a recent life event, watching a "guilty pleasure", high film ratings, and female gender. Medical conditions, age, income, alcohol use, and feeling tired or jetlagged were not significant.

**Conclusion**. People reporting the pseudo-phenomena of AALS are most likely experiencing "dramatically heightened exposure", watching as many films on a plane in a week's return trip as they would in a year at the cinema. Such perceptions are probably magnified by confirmation bias and further mentions in social media.

## INTRODUCTION

*Mark Kermode, Film critic: "Your movies…provoke very visceral, very physical reactions, very emotional responses and occasionally I think some critics are simply sideswiped by them".*

Corresponding author
Paul Wicks,
pwicks@patientslikeme.com

*David Cronenberg, Film director: "We should put all those critics on a plane and have them watch. I'm just noticing, and talking to some other people who've had the same reaction, that a movie that perhaps you would be very cool to on the ground, when you're watching on an airplane with the thrumming of the motors and the compressed air and so on, things suddenly become much more emotional and strange…A movie that I would probably have great disdain for on the ground, I find myself weeping over in an airplane. I don't know whether it's because you're constantly fearing for your life or what it is but seeing movies on an airplane is actually quite a different experience. A lesser one in some ways but weirdly more intense in others."*

(*Kermode & Mayo, 2007*)

Today, numerous anecdotal reports in mainstream and social media describe otherwise healthy individuals being prone to outbursts of uncontrollable crying while watching films on airplanes. An "index case" may be the film director David Cronenberg, who suggested that watching a film on an airplane caused him to experience a different set of emotional reactions than watching a film on the ground. This interview took place on the popular BBC podcast "Kermode and Mayo's Film Review", which has between 1–2 million listeners per week (*British Broadcasting Corporation, 2014*). Over the years, enthusiastic listeners to the program even coined a medical-sounding term for this potential phenomena: "altitude adjusted lachrymosity syndrome" (AALS). In AALS, healthy people report crying at scenes they wouldn't normally consider to be emotional when viewing films in other settings, such as at the cinema or at home, have difficulty suppressing the urge to cry, and are surprised at which films make them cry.

Since the potential index case, numerous claims of the phenomena have been made by prominent individuals in mass media settings. A subsequent sufferer, the film director Quentin Tarantino reported during a television interview on the Late Show with Stephen Colbert (around three million viewers): "There's something about watching a romantic comedy on an airplane flight; I think you become more emotional when you're three miles high in the air. I have found myself crying, just literally weeping at (movies I'd be embarrassed to confess that I'd watched)" (*Tarantino & Colbert, 2015*). A clip of the interview has been viewed on YouTube nearly 400,000 times. Another film director, Tim Burton, was quoted in the New York Times (with around nine million daily readers) as saying "I don't know why, maybe because you constantly think you're going to die—I find every movie, I cry if I watch it on a plane" (*Groening, 2014*). Another potential sufferer, the comedian David Baddiel said in an interview with the Guardian (around 150,000 daily readers): "I cry a lot. Especially on planes. What is that about? On a plane, I'll be in floods at The Good Dinosaur (an animated family film)" (*Greenstreet, 2016*). As a public health measure to educate passengers about this phenomena, in 2011 the airline Virgin Atlantic (flying around five million customers annually) placed "weep warning" disclaimers before films such as Toy Story 3, stating: "The following film contains scenes which may cause viewers of a sensitive disposition to cry, weep, sob, wail, howl, bawl, bleat or mewl (*Gunner, 2017*)" Such warnings were widely covered in the mainstream media and online including respectable sources like the BBC (*Gray, 2017*).

About two billion people fly commercially each year, with a recent review describing a range of potential associated health issues such as venous thromboembolism, jet lag, poor cabin air quality, cosmic-radiation exposure, and the exacerbation of pre-existing medical conditions (*Silverman & Gendreau, 2009*). Evidence from surveys, qualitative shadowing, and experience monitoring confirms that air travel can be a stressful experience, with stressors including going through security, getting to the gate on time, and being treated rudely by others (*Ahmadpour, Robert & Lindgaard, 2014*; *McMullin et al., 2014*).

Uncontrollable episodes of crying or laughter that are incongruent with an individual's feelings or situation affect as many as half of people living with neurological conditions such as ALS/MND, MS, stroke, or dementia (*Miller, Pratt & Schiffer, 2011*). Known variously as "labile affect", "emotional lability", "emotional incontinence", or pseudo-bulbar affect (PBA), disruption of glutamatergic cerebellar microcircuitry is thought to lead to motor disinhibition (*Ahmed & Simmons, 2013*). There is a single FDA-approved treatment for PBA, (dextromethorphan hydrobromide 20 mg + quinidine sulfate 10 mg, NueDexta®; Avanir Pharmaceuticals, Aliso Viejo, CA, USA) which can reduce the frequency of PBA attacks by around 50% (*Brooks et al., 2004*), and off-label use of antidepressants to treat PBA is common (*Ahmed & Simmons, 2013*). Consequences of PBA are serious and can involve embarrassment, social isolation, and depression as people around those affected find it hard to understand the affected individual's outbursts (*Strowd et al., 2010*).

The evidence for emotional changes in healthy people on airplanes is limited. A single randomized, controlled barometric manipulation study found a 4% decrease in oxygen saturation at altitude and evidence of altitude sickness symptoms including "distress" in healthy people subjected to a simulated 20-hour airline flight (*Muhm et al., 2007*).

However, with the exception of a single unscientific Facebook poll conducted by Virgin Atlantic, no researchers have sought to systematically understand the phenomenon and its risk factors. The development of new "diagnoses" can be harmful, leading to increased health anxiety, overdiagnosis, and unnecessary treatment (*Hurley, 2017*). The perpetuation of such myths in the media may be considered a manifestation of "fake news" in a medical context. Therefore, we sought to understand whether AALS was a real syndrome and, if so, to understand its underlying etiology.

## OBJECTIVES

Our prespecified hypothesis was that, controlling for other factors, participants watching a film in the air would be more likely to cry, feel the urge to cry, or be surprised at how much they cried than when watching a film on the ground. We also developed specific secondary hypotheses about potential risk factors identified by sufferers (described below).

## METHODS

A retrospective online survey was selected as the most practical approach, with an aim to recruit participants who had watched a film on an airplane and who had then subsequently watched another film on the ground.

## Survey development

Search terms on Pubmed such as "altitude adjusted lachrymosity syndrome", "crying on planes", "crying altitude", "altitude tearfulness", and "pseudobulbar affect altitude" failed to yield any previous studies. The use of online "grey literature" or social media for concept elicitation has been previously established as feasible (*McCarrier et al., 2016*), so a search on Google was performed for "Crying on planes", which produced a range of blogs, newspaper articles, and podcasts which anecdotally described the phenomenon in great detail (Appendix S1).

From these qualitative data, survey items (Appendix S2) were developed to estimate the likelihood of crying, uncontrollable crying, or surprise at crying, with items and responses based upon the validated Center for Neurologic Study Lability Scale (CNS-LS) (*Moore et al., 1997*). Participants were asked to categorize the film according to the categories listed on the Internet Movie Database (IMDB.com) and to rate the film for quality from 0–10 stars.

Further items evaluated the many speculative risk factors that have been reported by sufferers, such as genre of film (romantic comedies > dramas), choosing a film they would normally be embarrassed to watch (guilty pleasure > regular film choice), use of alcohol (alcohol > no alcohol), experiencing a recent emotional event (event > no event), or having a medical condition that might affect their emotional reaction such as a mood disorder or PBA-related condition like ALS/MND or MS (condition > no condition). Domains over which participants were unlikely to report reliable data, such as flight altitude, oxygen levels, air pressure, or more abstract concepts like "catharsis" were not given specific items, but an optional free text box was provided for any further observations about participants' emotions watching the film.

## Pilot fielding

A small pilot fielding was conducted using a US Census-representative sample via SurveyMonkey "Targeted Audience" system. To recruit a target $N$ of 50 who watched a film, 122 respondents were screened, with slight over-recruitment leading to $N = 56$.

## Minimizing bias

Amongst the pilot respondents, 33 (27%) had watched a film on a plane in the past 12 months, 49 (40%) more than 12 months ago, and 40 (33%) had never seen a film on a plane. Review of film titles showed that 35% of respondents who had last seen a film more than 12 months ago could not remember what they saw, compared to just 16% of those who had seen the film within the past 12 months. Accordingly, for the main study we changed the title of the survey invitation to clarify we were seeking responses only from respondents who had watched a film on a plane in the past 12 months to minimize recall bias. Median time to complete the test fielding was 3 min.

## Statistical analysis

Main hypotheses and exploratory factors were developed and defined in advance of data collection. Because respondents watched different films in each setting, a within-subjects comparison was not suitable. Due to the design of the survey and the release schedule of

films differing between ground viewing (summer blockbuster season) and plane viewing (typically a six month delay from the cinema), we planned for a binary logistic regression to predict the likelihood of crying at a film given 11 predictor variables: plane vs. ground location, film genre, gender, annual household income, film rating (0–10 stars, higher is better), experiencing a recent emotional life event, watching a guilty pleasure film, alcohol use, feeling tired, having a medical condition, and age (the only continuous variable). In addition, we report a number of unplanned exploratory post-hoc analyses using chi-square tests (alpha = 0.05; two tailed). Crying questions were asked using a Likert scale but collapsed into binary categories for analysis. Participants with missing data were excluded from further analysis. Statistical analysis were conducted in SPSS v21 (IBM Corporation, Armonk, NY, USA), *R Core Team (2017)* and *Python Software Foundation (2018)* using the Keras package.

## Power analysis

Given the absence of prior data from the literature we relied on pilot data to form a power analysis. From the pilot data, 30% of films watched on a plane elicited crying as opposed to 22% of films watched on the ground. Calculation of effect size using chi-squared statistic (*Hulley et al., 2013*) using a baseline risk of 22% and an exposed risk of 30% resulted in an odds ratio of 1.519, risk ratio of 1.364 and total required sample of 992 (496 reports on both ground and in air) to achieve 80% power (Type II error = 0.2) at a significance of 0.05 (Type I error rate).

## Survey population

While SurveyMonkey provided a representative sample for the pilot fielding, it was relatively expensive ($4.24 per complete survey, with $0.50 being donated to charity on each participant's' behalf). Therefore we sought to use Amazon Mechanical Turk (mTurk), a widely used resource for rapid gathering of voluntary research participants (e.g., a recent study on the potential health benefits of Pokemon Go (*Howe et al., 2016*)). While some concerns have been expressed about the validity and representativeness of mTurkers, a recent review of 35 validation studies in the health and medical literature confirmed "mTurk is an efficient, reliable, cost-effective tool for generating sample responses that are largely comparable to those collected via conventional meanings" (*Mortensen & Hughes, 2018*).

Guidance from the mTurk community itself suggests that "fair payment" is considered the US Federal minimum wage of $0.12/min (*Salehi, 2016*). Therefore respondents were offered $0.36 for completing the survey, which, with Amazon's fees, yielded a total cost per participant of $0.50. After a test run of $N = 9$ participants to ensure the survey was working and that data was valid (a co-pilot), the survey was fielded to a target $N$ of 1,000 mTurk workers. Only mTurk users living in the US with a 98% accuracy rating for prior tasks were eligible to take part. To minimize fraud, participants were provided with a random digit code at the end of the survey to enter into mTurk to receive their payment. Mechanical Turk users choose which studies to participate in and are free to withdraw at any time. As a minimal risk study with no personally identifiable information gathered, ethical approval was not sought for this project.

Per the Open Science Framework (osf.io), the authors confirm that they have reported all measures, conditions, data exclusions, and determination of sample size.

## RESULTS

Between July 14–July 16 2017, 1,238 participants started the survey of whom 154 were disqualified for having seen a film more than 12 months ago (12%) for a total of 1,084 surveys started. As others have reported (*Howe et al., 2016*), due to the nature of the mTurk platform, it is impossible to calculate a response rate from the number of people who may have seen the study posting online or understand the demographics of non-responders. Missing data was found for 49 respondents (4%) and these were excluded for a total of 1,035 respondents. The survey was closed for fielding on July 17th 2017 and took participants an average of 2.5 min to complete.

Amongst survey respondents, 450 (43.4%) were female, with a mean age of 32.1 years (SD 9.1, median 30, range 18–74) and with 37% living in households earning below $50,000 USD per year. Relative to 2015 US census data, this sample has fewer females (census: 50.8%), is younger (census median: 37.6 years), and is slightly wealthier than the general US population (*US Census Bureau, 2015*; *US Census Bureau, 2017*).

Respondents reported on 1,035 film viewings in the air and 904 on the ground, of which 340 (38%) were seen on television, 337 (38%) at the cinema, and 238 (26%) on a computer, tablet, or smartphone. Participants reported crying at 25% of films in the air (lower than the 30% identified in the pilot) as opposed to 22% on the ground (same as pilot). Table 1 describes the films reported, crying behaviour, and crying triggers. Examination of the frequencies clearly demonstrates that participants chose to watch different categories of films by setting, with more comedies, animated, and drama films being seen in the air and more action films being watched on the ground. There was not perfect agreement between participants on which genre a film belonged to, e.g., "Moana" could be classified as a family film, an animated film or a comedy. Where there was a modal response this was selected for classification, or in the event of a tie the film's category was selected from its listing on IMDB.com. Film genres were collapsed into larger groups on the basis of higher-order IMDB categories (e.g., comedy + romantic comedy).

For the primary analysis, a logistic regression model was fitted to the binary outcome of whether or not the participant cried at all (1) or never (0) while watching the film. Results are shown in Table 2.

Factors shown to be significant in predicting crying behaviour included the type of film (with dramas being much more likely to elicit crying), female gender, having had a recent emotional experience, choosing a "guilty pleasure" film, and watching a highly rated film (10/10 stars).

Contrary to our expectations, we did not find an effect of location (plane vs. ground) as significant in the model. Age, annual household income, feeling tired, or drinking alcohol were not significant factors either. Cross-validated predictive modelling using logistic regression resulted in a ROC AUC of 0.71, sensitivity 0.66 and specificity 0.64 when predicting probability of crying. Film category and gender dominated the model, consistent

**Table 1 Description of films, crying behaviour, and triggers.** Significance of unplanned post-hoc analyses shown for illustrative purposes only.

| | Plane (N = 1,035) | Ground (N = 915) | Sig |
|---|---|---|---|
| **Genre** | | | |
| Condensed film category | 15.0%-Comedy/RomCom<br>29.7%-Action/Thriller/Horror<br>12.1%-Adv/Fantasy/Sci/West<br>23.2%-Animated/Family<br>20.1%-Drama | 9.8%-Comedy/RomCom<br>48.8%-Action/Thriller/Horror<br>10.7%-Adv/Fantasy/Sci/West<br>13.3% - Animated/Family<br>17.4%-Drama | $X^2(4) = 84.049, p < .001$ |
| Most frequently viewed films (N, %) | 51 (4.9%)-Moana<br>35 (3.4%)-Rogue One<br>24 (2.3%)-King Kong: Skull Island<br>19 (1.8%)-La La Land<br>19 (1.8%)-Beauty and the Beast | 92 (10.2%)-Wonder Woman<br>68 (7.5%)-Spiderman: Homecoming<br>26 (2.9%)-Baby Driver<br>22 (2.4%)-Moana<br>21 (2.3%)-Despicable Me 3 | |
| **Crying Behaviour** | | | |
| How often did you find yourself crying very easily? | 74.8%-Never<br>16.8%-Rarely<br>6.1%-Occasionally<br>1.9%-Frequently<br>0.4%-Most of the time | 78.4%-Never<br>13.3%-Rarely<br>6.5%-Occasionally<br>1.5%-Frequently<br>0.2%-Most of the time | $X^2(4) = 5.800, p = .215$ |
| How often did you find it hard to control your urge to cry? | 74.3%-Never<br>15.9%-Rarely<br>6.4%-Occasionally<br>2.3%-Frequently<br>1.1%-Most of the time | 77.7%-Never<br>12.8%-Rarely<br>7.2%-Occasionally<br>1.7%-Frequently<br>0.7%-Most of the time | $X^2(4) = 6.330, p = .176$ |
| Were you surprised at how much you cried or felt like crying at this film? | 8.6%-Yes<br>91.4%-No | 11.3%-Yes<br>88.7%-No | $X^2(1) = 3.915, p = .056$ |
| **Potential Crying Triggers** | | | |
| A recent emotional experience | 28.5%-Yes<br>71.5%-No | 19.2%-Yes<br>80.8%-No | $X^2(1) = 22.538, p < 0.001$ |
| Film was a "guilty pleasure" | 13.3%-Yes<br>86.7%-No | 9.1%-Yes<br>90.9%-No | $X^2(1) = 8.716, p = 0.002$ |
| Alcohol consumed | 18.5%-Yes<br>81.5%-No | 14.2%-Yes<br>85.8%-No | $X^2(1) = 6.475, p = 0.006$ |
| Feeling tired / jetlagged | 42.9%-Yes<br>57.1%-No | 4.8%-Yes<br>95.2%-No | $X^2(1) = 373.247, p \leq 0.001$ |
| Medical condition | 1.3%-Yes<br>98.7%-No | 0.8%-Yes<br>99.2%-No | $X^2(1) = 1.097, p = 0.206$ |

with the findings above. Similar results were seen when predicting the urge to cry, and the feeling of being surprised by crying. Contrary to our expectations, more participants reported being "surprised at crying" at films on the ground (11.3%) than in the air (8.6%).

In clinical practice, air travellers may exhibit multiple risk factors; therefore Table 3 outlines a stepped treatment approach modelled on guidance for the prevention of deep vein thrombosis in the air (*Silverman & Gendreau, 2009*). Table 4 identifies individual films current at the time of the study but also particular genres that are likely to induce crying.

**Table 2  Results of the primary regression for likelihood of crying in descending order of statistical significance.**

| Coefficients: | Estimate | Std. error | z value | Pr(> |z|) | Sig |
|---|---|---|---|---|---|
| (Intercept) | −2.474423 | 0.923096 | −2.681 | 0.00735 | ** |
| Significant predictors | | | | | |
| QGender2 (Female gender) | 0.765398 | 0.125998 | 6.075 | 1.24E−09 | *** |
| RecodedFilmCat6 (Drama genre) | 1.37876 | 0.236487 | 5.83 | 5.54E−09 | *** |
| TrigEmotion2 (Life event) | −0.442539 | 0.138248 | −3.201 | 0.00137 | ** |
| RecodedFilmCat3 (Adv/etc. genre) | 0.736506 | 0.2679 | 2.749 | 0.00597 | ** |
| RecodedFilmCat4 (Ani/fam genre) | 0.658505 | 0.242406 | 2.717 | 0.0066 | ** |
| TrigGuilty2 (Guilty pleasure) | −0.457585 | 0.20214 | −2.264 | 0.02359 | * |
| FilmRate10 (10 stars) | 1.705725 | 0.797669 | 2.138 | 0.03249 | * |
| Non-significant predictors | | | | | |
| FilmRate9 (9 stars) | 1.368903 | 0.798016 | 1.715 | 0.08627 | . |
| FilmRate8 (8 stars) | 1.187144 | 0.798658 | 1.486 | 0.13717 | |
| QIncome10 ($175k–200k) | −0.648126 | 0.555446 | −1.167 | 0.24327 | |
| QIncome4 ($25k–50k) | −0.365927 | 0.349919 | −1.046 | 0.29568 | |
| TrigAlco2 (Had alcohol) | 0.146756 | 0.1729 | 0.849 | 0.396 | |
| FilmRate7 (7 stars) | 0.63025 | 0.806869 | 0.781 | 0.43474 | |
| RecodeLocation1 (On a plane) | 0.102195 | 0.140273 | 0.729 | 0.46628 | |
| QIncome7 ($100k–125k) | −0.277864 | 0.383706 | −0.724 | 0.46897 | |
| FilmRate6 (6 stars) | 0.558507 | 0.826638 | 0.676 | 0.49927 | |
| QIncome3 ($10k–25k) | 0.256009 | 0.379641 | 0.674 | 0.50009 | |
| QIncome5 ($50k–75k) | −0.204009 | 0.347717 | −0.587 | 0.5574 | |
| QIncome9 ($150k–175k) | −0.243521 | 0.449845 | −0.541 | 0.58827 | |
| QIncome6 ($75k–100k) | −0.178787 | 0.357963 | −0.499 | 0.61746 | |
| TrigTired2 (Tired / jetlagged) | 0.070974 | 0.159659 | 0.445 | 0.65666 | |
| QAge | −0.002321 | 0.00653 | −0.355 | 0.7223 | |
| FilmRate5 (5 stars) | −0.261219 | 0.885067 | −0.295 | 0.76789 | |
| FilmRate3 (3 stars) | −0.352021 | 1.305904 | −0.27 | 0.7875 | |
| QIncome2 ($0–10k) | −0.126222 | 0.51853 | −0.243 | 0.80768 | |
| QIncome11 ($200k+) | −0.116256 | 0.531427 | −0.219 | 0.82683 | |
| RecodedFilmCat2 (Action/etc.) | −0.043251 | 0.23609 | −0.183 | 0.85465 | |
| QIncome8 ($125k–150k) | 0.037557 | 0.445115 | 0.084 | 0.93276 | |
| FilmRate4 (4 stars) | −0.055034 | 0.893157 | −0.062 | 0.95087 | |
| FilmRate2 (2 stars) | −13.24816 | 309.724689 | −0.043 | 0.96588 | |

**Notes**
*** $p < 0.001$.
** $p < 0.01$.
* $p < 0.05$.
· $p < 0.1$.

# DISCUSSION

We found that AALS is probably not a real medical condition. We found no evidence of increased crying resulting from high altitude in terms of frequency of crying, control over crying, or surprise at crying at a heterogeneous array of films within the same

**Table 3** Risk factors for crying in a hypothetical series of air travellers and recommended treatment approaches (modelled on *Silverman & Gendreau, 2009*) risk factors for in-flight thrombosis.

| | Definition | Recommendation |
|---|---|---|
| Low risk (~25% risk of crying) | Watching a mediocre action film while traveling on business. | Avoid listening at excessive volume on headphones. |
| Moderate risk (~50% of crying) | Watching a pair of high quality "guilty pleasure" animated films. | Maintain hydration, bring soft paper tissues or handkerchief. |
| High risk (~100% risk of crying) | Watching four consecutive Oscar-winning dramas about historical figures overcoming adversity while traveling to a funeral. | Let it all out. |

**Table 4** Films and genres identified as the highest risk factors for crying while watching.

| Films |
|---|
| 1. The ZooKeeper's Wife (100% risk of crying, $N = 8$, Drama) |
| 2. Okja (90%, $N = 10$, Science Fiction) |
| 3. Lion (65%, $N = 17$, Drama) |
| 4. La La Land (59%, $N = 22$, Drama) |
| 5. Moana (42%, $N = 73$, Animated/Family) |
| 6. Beauty and the Beast (40%, $N = 30$, Animated/Family) |
| 7. Hidden Figures (39%, $N = 18$, Drama) |
| 8. Zootopia (38%, $N = 13$, Animated/Family) |
| 9. Arrival (38%, $N = 16$, Science Fiction) |
| 10. The Martian (36%, $N = 11$, Science Fiction) |

| Genres |
|---|
| 1. 42.5%—Drama (including biography, historical, romance) |
| 2. 28.9%—Animated/Family |
| 3. 25.2%—Adventure/Fantasy/Science Fiction/Westerns |
| 4. 14.4%—Action/Thriller/Horror |
| 5. 13.5%—Comedy/Romantic Comedy |

cohort of individuals on planes as opposed to on the ground. This seems true despite significantly higher levels of proposed triggers for AALS in the air, such as experiencing a recent emotional event, feeling tired or jetlagged, drinking alcohol, or watching a "guilty pleasure" film. Contrary to the qualitative descriptions identified in the grey literature review, there did not appear to be an elevated degree of crying at comedies or romantic comedies on planes, which had the lowest rates of crying overall. Reassuringly for medical professionals, there do not appear to be any major implications for aviation medicine in that only a handful of participants endorsed medical conditions such as depression, anxiety, pregnancy, or multiple sclerosis as worsening their emotional reactions to a film. Future studies could replicate these findings in a group of patients at greater risk of pseudobulbar affect by inviting participants to watch films like in different settings, with and without medication.

The most likely explanation for the perceived phenomenon of AALS is *dramatically heightened* exposure to films in a short space of time. On long-haul flights passengers watch an average of three films each (Anonymous airline, pers. comm., 2017) and, assuming a

return flight, this could mean exposure to six films within a week. With an unadjusted crying rate of about one in four, that would deem any passenger almost certain to cry at least once at a film on a plane in the course of their trip. By contrast, data from the US National Association of Theatre Owners (NATO) suggests an average of five trips to the cinema per annum (*National Association of Theatre Owners NATO, 2015*). With about half of these viewings being action films unlikely to elicit crying, it is clear how the perceived phenomena may have taken root. This, combined with repeated descriptions in the media and cognitive heuristics such as confirmation bias probably account for the reported pseudo-phenomenon.

Other factors that were identified by the model included watching dramas or family films, "guilty pleasure" films, and high-quality films, and having recently experienced an emotional life event. While a review of factors implicated in adult crying is outside the scope of the current study (see *Vingerhoets & Bylsma, 2016*), such factors may be more associated with "sentimental or moral tears shed when watching movies with themes such as eternal love, self-sacrifice, altruism, the good that overcomes the bad, etc." While gender arose as a significant variable in the model, this is likely to have complex underpinnings including socialization to certain film choices, and that males may be less likely to accurately self-report instances of crying (*Fischer & LaFrance, 2015*). We were surprised at the lack of surprise reported by participants at crying on planes, given that this appeared to be a signature characteristic of AALS. However, this may have to do with expectations for the different film choices made and any pre-existing belief in something like the AALS pseudo-phenomena i.e., they were not surprised because they chose sad films on a plane. We were also surprised at the relatively low rates of alcohol use in-flight relative to ground settings, which may reflect a move by airlines away from free distribution of alcohol to a cash bar, at least in economy class.

The screening of feature-length films on aircraft first became common in the 1960's, and in addition to the commercial and marketing benefits offered the potential to calm nervous flyers anxious about long-haul flights (*Groening, 2016*). Originating as 16 mm film projected to drop-down cinema screens at the front of the cabin, IFE systems evolved into drop-down televisions in the aisles, then seat-back screens in the late 1980's and to modern "audio-visual on demand" (AVOD) touchscreens from the early 2000's onwards (*Mele, 2017*). As technology has improved, passengers have been getting closer and closer to their screens, gained more control over the choices available to them, and through increasingly sophisticated noise-cancelling headphones have been drawn into more of a "bubble" in which they can be distracted from the noise and stress of their flight and from other passengers (*Groening, 2016*). One model proposes watching IFEs as a "mediator" of emotional function during flight for helping to focus passengers' attention away from distressing elements such as the cramped conditions, troublesome neighbours, or boredom (*Ahmadpour, Robert & Lindgaard, 2014*). In support of passenger film choice as the key mechanism, reports of AALS in popular media do not predate the "video on demand" era, i.e., when whole planeloads of passengers were watching the same film either projected at the front of the plane, or shown in a fixed loop on seat-back screens, there were no reports twenty years ago of mass hysterical crying at the end of "Titanic", for instance.

The pseudo-phenomenon of AALS is likely to be time-limited, as commercial airlines are moving from today's expensive, cumbersome, and heavy AVOD IFE systems to much cheaper methods of providing high speed internet access to passengers (*Mele, 2017*). As passengers migrate from watching films to streaming live television, playing games, participating in social media, or working, we might expect to see a decline in mentions of AALS, which could be confirmed through tools such as Google Trends.

One potentially interesting finding arising in our review of the grey literature was that many newspaper articles and blog posts (Appendix S1) such as Vice, Bustle, Junkee, New Statesman, and the Guardian, posed a question which the article itself didn't answer, such as "Why do we always cry on planes?" Such headline techniques have been described as "Clickbait" designed solely to garner online views to gain advertising revenue (*Khoja, 2016*). Further review of the article text showed a high degree of overlap in content (sometimes verging on plagiarism) and heavy reliance as a reference on a single unscientific poll commissioned by Virgin Atlantic. In an age of "fake news" it is likely that AALS is merely a "pop psychology" urban legend, kept alive by confirmation bias when an individual watches six films on a plane, cries at one, and then selectively recalls only the experience of crying on a plane when reading a prompt in the news media before going on to "share" the story on social media to further perpetuate the meme.

## Limitations

This study had a number of limitations that may compromise validity such as selection bias, recall bias, the heterogeneity of viewing experiences in the air (e.g., long-haul vacation with children in economy vs. traveling alone for work in business class) as well as on the ground (e.g., watching a rerun on television vs. going to the cinema to watch a new release). Researchers have cautioned that results from mTurk may not generalize to the US population, and we found our respondents were more likely to be male, younger, and wealthier than average citizens (*Mortensen & Hughes, 2018*) Seasonality was not controlled for, which would have allowed us to discount seasonal affective disorder; however data was gathered in the summer and most cinema viewings appeared to be recent releases. We did not gather information on which scenes prompted crying or whether it was the first time a participant had watched this film. We did not screen participants for psychiatric disorders with a validated measure, which could have improved confidence that mood disorders were not a significant factor. We did not ask participants if they were taking any medication, which may have been a factor.

While the ideal experimental design would involve a randomized crossover trial with participants watching the same film in the air and on the ground (perhaps in a flight simulator) and having their lachrymosity recorded using Schirmer's test (*Schirmer, 1903*), or perhaps a thermal imaging camera (*Ioannou et al., 2016*), this was deemed impractical and rather expensive. Therefore, this study may be considered a precursor to a more robust experimental design.

## CONCLUSIONS

In an era of "fake news", "too much medicine" and the "replication crisis" of the social sciences, it is important that we do not take widely-accepted phenomena as statements of fact. The cognitive biases and heuristics that allow us to function routinely in the world are poor proxies for understanding the truth of complex low-frequency events. The rapid deployment of online surveys to large anonymous samples may be one tool to help dispel urban legends and encourage a more critical approach to human behaviour.

## ACKNOWLEDGEMENTS

The authors are grateful to the participants who took part in this study, for advice and insights from Jovita Toh of Encore Inflight Ltd, Simon Cuthbert of Cathay Pacific, and an airline inflight entertainment manager who wished to remain anonymous. The authors would also like to say hello to Jason Isaacs.

### Funding
The authors received no funding for this work.

### Competing Interests
PW is an employee of PatientsLikeMe and holds stock options in the company. LL is an employee of Cohens Veterans Biosciences.

PW is an associate editor at the Journal of Medical Internet Research and is on the Editorial Boards of The BMJ, BMC Medicine, and Digital Biomarkers. The PatientsLikeMe Research Team has received research funding (including conference support and consulting fees) from Abbvie, Accorda, Actelion, Alexion, Amgen, AstraZeneca, Avanir, Biogen, Boehringer Ingelheim, Celgene, EMD, Genentech, Genzyme, Janssen, Johnson & Johnson, Merck, Neuraltus, Novartis, Otsuka, Permobil, Pfizer, Sanofi, Shire, Takeda, Teva, and UCB. The PatientsLikeMe R&D team has received research grant funding from Kaiser Permanente, the Robert Wood Johnson Foundation, Sage Bionetworks, The AKU Society, and the University of Maryland. PW has received speaker fees from Bayer and honoraria from Roche, ARISLA, AMIA, IMI, PSI, and the BMJ.

### Author Contributions
- Paul Wicks conceived and designed the experiments, performed the experiments, analyzed the data, contributed reagents/materials/analysis tools, prepared figures and/or tables, authored or reviewed drafts of the paper, approved the final draft.
- Lee Lancashire analyzed the data, contributed reagents/materials/analysis tools, prepared figures and/or tables, authored or reviewed drafts of the paper, approved the final draft.

### Human Ethics
The following information was supplied relating to ethical approvals (i.e., approving body and any reference numbers):

As a minimal risk study with no personally identifiable information gathered, ethical approval was not sought for this project.

This exemption is consistent with US Code of Federal Regulations

Exemption 45 CFR 46.101(b)(2) or (b)(3): https://www.hhs.gov/ohrp/regulations-and-policy/decision-trees-text-version/index.html#ch04.

### Data Availability

Wicks, Paul; Lancashire, Lee (2018): No tears in heaven—Altitude adjusted lachrymosity syndrome—Results from an online survey. figshare. https://doi.org/10.6084/m9.figshare.5783508.v1.

### Supplemental Information

Supplemental information for this article can be found online at http://dx.doi.org/10.7717/peerj.4569#supplemental-information.

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
