# Peer review of "No tears in heaven: did the media create the pseudo-phenomenon "altitude-adjusted lachrymosity syndrome (AALS)"?"

_PeerJ, doi:10.7717/peerj.4569_

## Round 0.1 · original submission · Major Revisions

The reviewers like your approach. Please address all concerns and also provide a statement in your resubmission cover letter that this research is veritable and not a hoax (this is a reviewer concern, given the tongue-in-cheek handling of table 3, etc.). I’m ok with it: but put your money where your mouth is so that readers do not write off the work.

Reviewer 1 ·

Basic reporting

It's fine.

Experimental design

It's fine.

Validity of the findings

Legit

Additional comments

“Tears in heaven” by Wicks & Lancashire takes a closer look at “AALS”.
The paper is interesting and quirky. It will be sure to draw media attention.
The style of the paper is tongue in cheek throughout. This is most noticeable at the end of the conclusions: “Therefore this study may be considered a precursor to
a more robust experimental design.”. Sure. Also, table 3. This is not necessarily a problem, as most of the comparable literature is dry as dust and not conducive to anyone actually reading it. In that sense, this was quite refreshing.

Assuming that the authors are actually serious, here is my review.

Strengths:
*Most people ask turkers to do too much, specifically for too long. There is only so much a turker will do. The authors to not make this mistake, as they keep their survey at much less than 5 minutes (which I consider to be the max a Turker will do without resorting to cheating).
*The authors provide a plausible mechanism as to why the phenomenon might have gained traction. It is fairly obvious what is going on. People watch more movies that tend to elicit crying (dramas and family movies) on planes (probably due to the available selection) and they are exposed to more movies on a single trip than during an entire year on the ground, with selective remembering and confirmation bias doing the rest.
*The authors are willing to report a “non-result”. This is very important and should be supported.
*The paper is well written. This is rare enough to note it here.

Major concerns:
1. Are the authors fighting a strawman? Most of the people cited in the background are directors and movie people. I was entirely unaware of an impending epidemic of AALS, didn’t even know it is a thing. I fly a lot, but have never seen anyone cry on a plane (of course, I have seen many people watch movies on planes, including myself). As the authors state themselves, there are no academic studies on this. Their own pubmed search came up empty. Then again, it is important to dispel myths and urban legends before they gain much traction. Then again, why does it matter what a couple of neckbeards in their basement posting on the internet believe about who cries on planes? The authors would be well advised to google the “Mandela effect”. Thousands (millions?) of people online are willing to believe that it is more likely that the Illuminati changed the timeline of history rather than questioning the perfect veracity of their memory.
2. As the authors do report a non-result, adequate power is critical. I strongly advise the authors to elaborate and clarify their power analysis. I must confess that I do not understand it. As far as I can tell, no suitable power analysis was performed. This is particularly important as what the authors report relies on the pilot data, which might misrepresent the true effect size. This is true both for the section on power, as well as anywhere else power is mentioned in the manuscript, e.g. when talking about the Learning curve analysis. I did not find this compelling. Please clarify/elaborate. Moreover, 1,000 participants sounds like a lot, but it really isn’t. In this context, it is important to note that Turkers are not actually real people in this sense. There are all kinds of problems with Turker samples, see for instance https://osf.io/cv2bn/

Minor concerns:
1. I was surprised that turkers fly so much. And saw a movie on the plane – so were presumably flying long distance. Is this plausible, given that they work for minimum wage on the internet (less than other jobs in the gig economy, e.g. Uber drivers)? In other words: Do turkers fly (this much)? A priori, I would have expected Turkers to be the least like sample to do so, and thus the least suitable to address this study. Maybe I misunderstood what the authors report, but I don’t think so.
2. Please clarify the modeling of the crying behavior. The text (and the model) implies that crying was binary (cried or not), but the table suggests otherwise.
3. 219: (Saheli) is Saheli 2017, no?
4. 264: Learning curve analysis? This comes out of nowhere. Elaborate.
5. In general, put a space before an opening parenthesis. They seem to be missing throughout. Which makes me suspect the authors did this on purpose.
6. Title suggestion: Shouldn’t it be “no tears in heaven”?
7. The authors keep talking about their deep net. But this doesn't really seem to be well motivated at all. Or integrated into the rest of the manuscripts. All results are in terms of the logistical regression. What happened to the deep net? Why mention it at all, if it is not used?

·

Basic reporting

The review of relevant literature was thorough and well-organized. The authors situated the current research well within the "era of fake news," clearly articulated the importance of evaluating claims that have become commonplace in the popular media. I have no substantive suggestions for changes. My minor critiques are as follows:

• There appears to be a missing citation on line 77.
• I do not have experience using Mechanical Turk. In the shared data file, could “RespondentID” numbers be used to identify specific users of M-Turk, or are these ID numbers generated by the researchers? I just want to be sure that the data have been adequately anonymized.

Experimental design

I request that the authors add a statement to the paper confirming whether they have reported all measures, conditions, data exclusions, and how they determined their sample sizes. The authors should, of course, add any additional text to ensure the statement is accurate. This is the standard reviewer disclosure request endorsed by the Center for Open Science [see http://osf.io/project/hadz3]. I include it in every review.

Validity of the findings

Although I had a good laugh while reading Table 3, I question whether its content has a solid empirical foundation and whether it adds substantive value to this paper. I will leave it to the handling editor to decide.

Additional comments

This manuscript was a pleasure to read. I literally laughed out loud at a few points. Laughter while reviewing can be a death knell for a manuscript, but it was not in this case. The authors provided valuable (and scientifically sound) myth busting and did so in an engaging manner that will be appealing to the readership of PeerJ. My critiques are minimal and should be easily addressed.

---

## Round 0.2 · accepted · Accept

The reviewers have read your revised manuscript and are pleased with your revisions.

# Reviewer 1 ·

Basic reporting

Good

Experimental design

Good

Validity of the findings

Good

Additional comments

Thanks for addressing my concerns and taking my suggestions seriously. I think the manuscript is actually improved. So... good job?

·

Basic reporting

no comment

Experimental design

no comment

Validity of the findings

no comment

Additional comments

I am satisfied that all of my concerns have been addressed. Kudos on a fun and interesting myth-busting exercise.